# Evaluation of 6-OxP-CD, an Oxime-based cyclodextrin as a viable medical countermeasure against nerve agent poisoning: Experimental and molecular dynamic simulation studies on its inclusion complexes with cyclosarin, soman and VX

Edmond Y. Lau[1]*, Heather A. Enright[1], Victoria Lao[1], Michael A. Malfatti[1], Brian P. Mayer[2,3,4], Audrey M. Williams[2,3,4], Carlos A. Valdez[2,3,4]*

1 Biosciences and Biotechnology Division, Lawrence Livermore National Laboratory, Livermore, CA, United States of America, 2 Global Security Directorate, Lawrence Livermore National Laboratory, Livermore, CA, United States of America, 3 Nuclear and Chemical Sciences Division, Lawrence Livermore National Laboratory, Livermore, CA, United States of America, 4 Forensic Science Center, Lawrence Livermore National Laboratory, Livermore, CA, United States of America

* lau12@llnl.gov (EYL); valdez11@llnl.gov (CAV)

## Abstract

The ability of the cyclodextrin-oxime construct 6-OxP-CD to bind and degrade the nerve agents Cyclosarin (GF), Soman (GD) and S-[2-[Di(propan-2-yl)amino]ethyl] O-ethyl methyl-phosphonothioate (VX) has been studied using $^{31}$P-nuclear magnetic resonance (NMR) under physiological conditions. While 6-OxP-CD was found to degrade GF instantaneously under these conditions, it was found to form an inclusion complex with GD and significantly improve its degradation ($t_{1/2}$ ~ 2 hrs) relative over background ($t_{1/2}$ ~ 22 hrs). Consequently, effective formation of the 6-OxP-CD:GD inclusion complex results in the immediate neutralization of GD and thus preventing it from inhibiting its biological target. In contrast, NMR experiments did not find evidence for an inclusion complex between 6-OxP-CD and VX, and the agent's degradation profile was identical to that of background degradation ($t_{1/2}$ ~ 24 hrs). As a complement to this experimental work, molecular dynamics (MD) simulations coupled with Molecular Mechanics-Generalized Born Surface Area (MM-GBSA) calculations have been applied to the study of inclusion complexes between 6-OxP-CD and the three nerve agents. These studies provide data that informs the understanding of the different degradative interactions exhibited by 6-OxP-CD with each nerve agent as it is introduced in the CD cavity in two different orientations (up and down). For its complex with GF, it was found that the oxime in 6-OxP-CD lies in very close proximity ($P_{GF}\cdots O_{Oxime}$ ~ 4–5 Å) to the phosphorus center of GF in the 'down$_{GF}$' orientation for most of the simulation accurately describing the ability of 6-OxP-CD to degrade this nerve agent rapidly and efficiently. Further computational studies involving the center of masses (COMs) for both components (GF and 6-OxP-CD) also provided some insight on the nature of this inclusion complex. Distances between the COMs (ΔCOM) lie closer in space in the 'down$_{GF}$' orientation than in the

**Data Availability Statement:** All relevant data are within the paper and its Supporting Information files.

**Funding:** This work was performed under the auspices of the U. S. Department of Energy by Lawrence Livermore National Laboratory under Contract DE-AC52-07NA27344. The work was funded by a grant from the Defense and Threat Reduction Agency (DTRA) to C. A. V. (Grant number: CB10902). The funders had no role in study design, data collection and analysis, decision to publish, or preparation of the manuscript.

**Competing interests:** The authors have declared that no competing interests exist.

'up$_{GF}$' orientation; a correlation that seems to hold true not only for GF but also for its congener, GD. In the case of GD, calculations for the 'down$_{GD}$' orientation showed that the oxime functional group in 6-OxP-CD although lying in close proximity (P$_{GD}$···O$_{Oxime}$ ~ 4–5 Å) to the phosphorus center of the nerve agent for most of the simulation, adopts another stable conformation that increase this distance to ~ 12–14 Å, thus explaining the ability of 6-OxP-CD to bind and degrade GD but with less efficiency as observed experimentally (t$_{1/2}$ ~ 4 hr. vs. immediate). Lastly, studies on the VX:6-OxP-CD system demonstrated that VX does not form a stable inclusion complex with the oxime-bearing cyclodextrin and as such does not interact in a way that is conducive to an accelerated degradation scenario. Collectively, these studies serve as a basic platform from which the development of new cyclodextrin scaffolds based on 6-OxP-CD can be designed in the development of medical countermeasures against these highly toxic chemical warfare agents.

## Introduction

Recent events such as the assassination attempts on Sergei and his daughter Yulia Skripal in the UK [1] and on Alexei Navalny in Russia [2] have launched nerve agents into the spotlight yet again, spreading fear and concern around the world [3]. The use of organophosphorus-based nerve agents, comprised by the traditional agents belonging to the G- and the V-series and the newly introduced Novichoks, under any circumstances is banned by the chemical weapons convention (CWC) [4]. However, this world-wide agreement has been ignored numerous times in the past forty years with some of the most deplorable instances involving the use of sarin gas during the Iran-Iraq conflict [5], and again during the chemical attacks in Ghouta [6] and Khan Shaykun [7] in Syria to the assassination of Kim Jong-Nam at the Kuala Lumpur airport in Malaysia involving the nerve agent VX [8] (Fig 1A). Due to their lethal effects, directly arising from their inhibitory actions on the enzyme acetylcholinesterase (AChE) [9, 10], these chemicals have been a focal point of a multipronged research effort around the globe centering in the development of technologies that deal with their analytical detection [11–13], destruction [14, 15], protection against [16, 17] and antidote development [18–20]. As an integral part of the latter effort, methods of treatment rely heavily on the use of quaternary oxime salts that serve to reactivate any inhibited AChE in conjunction with a muscarinic receptor antagonist such as atropine that are coadministered via an intramuscular (IM) autoinjector known as the Duodote [20] (Fig 1B). Two of the most used quaternary oxime reactivators include 2-PAM [21] and MMB4 [22], although recent research efforts have been geared towards neutral oximes that can provide protection not only to the peripheral but the central nervous system such as RS194B [23, 24] and LLNL-02 [25]. Another similar line of research deals with the development of prophylactics for use as medical countermeasures (MCMs) [26]. The goal of a prophylactic treatment is to provide a layer of protection to AChE prior to and during the exposure to a nerve agent. The ideal prophylactic should possess characteristics such as high selectivity for the nerve agent, rapid kinetics for agent degradation and a long circulating half-life. An approach that has become a center of intense research among groups involved in this area, is the use of supramolecular scaffolds that can readily circulate in the bloodstream and once exposure to the nerve agent occurs, the nerve agent is trapped in their interior and thus successfully neutralized [27].

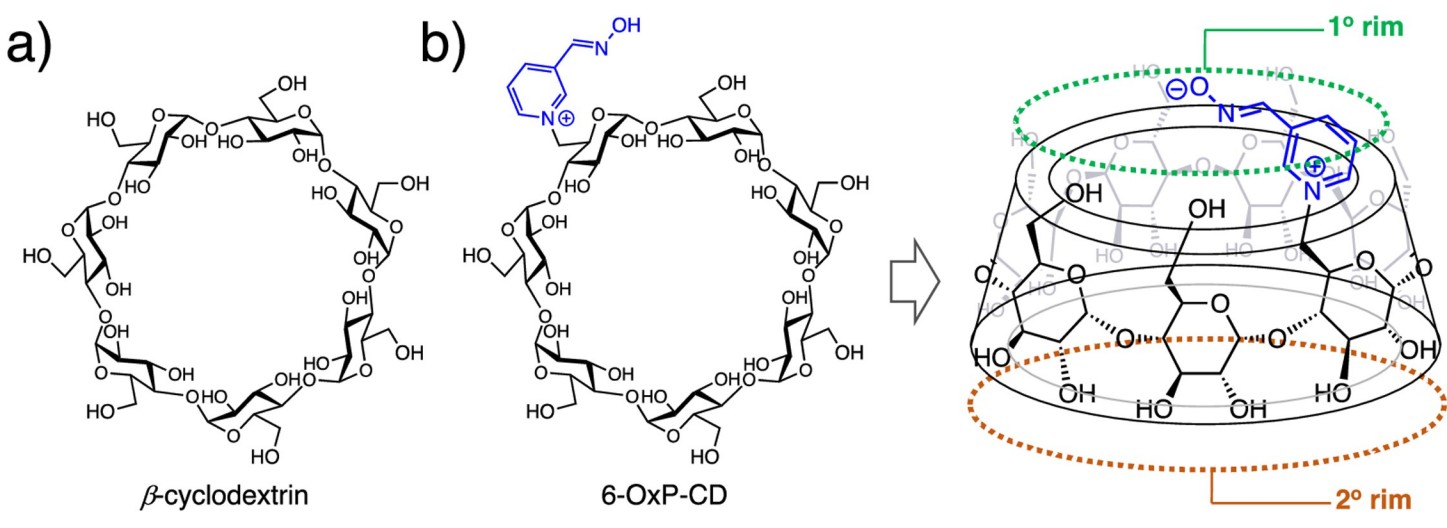

**Fig 1.**

To this end, an interesting family of supramolecular scaffolds that have been heavily explored to perform this prophylactic function are the cyclodextrins [28–30]. Cyclodextrins (CDs) are cyclic oligosaccharides that are composed of glucosyl units linked in an α-1,4-fashion (Fig 2A) [28]. It is this arrangement of glucosyl units that gives CDs their characteristic three-dimensional structure resembling that of a frustrum or a cone that is open at both ends, one larger than the other. An interesting property of CDs is that their interior is hydrophobic in nature while their exterior is hydrophilic, therefore allowing these scaffolds to serve as hosts to a myriad of organic molecules [31–33] in an aqueous environment, including those belonging to the nerve agent class (Fig 2B) [34–36]. CDs have become molecules of interest due to their established use in the medicinal field coupled to the various methods available for their

**Fig 2.**

chemical modification for specific applications [37]. Their potential use as MCM entities against nerve agents was realized in 1986 when studies by Désiré and Saint-André demonstrated their binding ability and neutralization power against Sarin (GB) and Soman (GD) [38, 39]. This unique property to form such inclusion complexes in aqueous media was used in 2011 to generate a modified *β*-CD, named 6-(3-oximepyridine)-CD or 6-OxP-CD, that could bind and degrade the nerve agent cyclosarin (GF) (Fig 2B) [40]. The report by the Kubik group at the Technische Universität Kaiserslautern in collaboration with the Bundeswehr Institute of Pharmacology and Therapy (BIPT), both in Germany, demonstrated that 6-OxP-CD protected AChE from the effects of GF *in vitro*. Subsequent *in vivo* studies using a guinea pig model, demonstrated some level of protection given by 6-OxP-CD to these animals [41]. Interestingly, in their original report the authors also observed that native *β*-CD provided a decent degree of protection to AChE but not as potent as 6-OxP-CD [40]. Motivated by the results obtained in these studies, our group conducted further testing on 6-OxP-CD's ability to bind and degrade two additional nerve agents, Soman (GD) and *S*-[2-[Di(propan-2-yl)amino] ethyl] *O*-ethyl methylphosphonothioate (VX). In addition, molecular dynamics (MD) simulations were carried out to gain an understanding on the nature of the inclusion complexes and hydrolytic profile between 6-OxP-CD and all three nerve agents separately.

## Methods

### Materials

All chemicals were purchased from commercial suppliers and used as received. 3-pyridine aldoxime (3-PA), *p*-toluenesulfonyl chloride (tosyl chloride), β-CD, were purchased from Sigma-Aldrich (St. Louis, MO.). Sodium bicarbonate and anhydrous sodium sulfate were purchased from Acros Organics (Westchester, PA.). Deuterium oxide ($D_2O$) and sodium carbonate were purchased from Alfa Aesar (Ward Hill, MA). 6-deoxy-6-tosyl-β-CD was synthesized and purified as described in the literature [42]. 6-OxP-CD was synthesized as described by Zengerle et al. [40] and purified by semi-preparative HPLC and lyophilized using a Labconco FreeZone 4.5 Liter Lyophilizer (-50 °C). Acrodisc PTFE syringe filters (0.45 μm) were purchased from Pall laboratories (Port Washington, NY.). Autosampler vials and glass inserts were purchased from Agilent Technologies (Santa Clara, CA.). Solvents used during the syntheses were removed by using a Büchi rotary evaporator R-200 equipped with a Büchi heating bath B-490 and coupled to a KNF Laboport Neuberger UN820 vacuum pump. Analytical thin layer chromatography (TLC) was conducted on Agela Technologies silica gel glass plates ($AcOH/CHCl_3/H_2O$, 8:1:1) [43] coupled with detection by ceric ammonium molybdate (CAM) [44–46], exposure to iodine vapor and/or UV light ($\lambda$ = 254 nm) [47–49]. HRMS analyses were obtained at the Forensic Science Center at the Lawrence Livermore National Laboratory using Chemical Ionization (CI).

### Nuclear magnetic resonance

Spectra were obtained using a Bruker Avance III 600 MHz instrument equipped with a Bruker QNP 5 mm cryoprobe (Bruker Biospin, Billerica, MA) at 30.0 ± 0.1˚C. NMR data is reported as follows: chemical shift ($\delta$) (parts per million, ppm); multiplicity: m (multiplet), d (doublet), t (triplet), q (quartet), app t (apparent triplet), tt (triplet of triplets), qd (quartet of doublets), sep (septet); coupling constants (*J*) are given in Hertz (Hz). ${}^1$H NMR (600 MHz) chemical shifts are calibrated with respect to residual HOD in $D_2O$ centered at 4.75 ppm. ${}^{13}$C-DEPT-135 NMR was used to identify the nature (*i.e.* $1^o$, $2^o$, $3^o$ or quaternary) of the carbon atoms in the synthesized targets.

## Chemical synthesis

6-OxP-CD was synthesized as described by Zengerle et al. [40]. The final product was purified by semi-preparative high performance liquid chromatography using acetonitrile:water as the mobile phase. Fractions were collected using the mass filter collection mode as well as a UV detector and lyophilized prior to its use. All analytical data ($^1$H, $^{13}$C, $^{13}$C-DEPT-135 NMR and HRMS) on 6-OxP-CD matched those previously published for the material [40].

**Caution notice.** Nerve agents are highly toxic compounds that can harm exposed individuals at extremely small doses. Only properly trained personnel, in a certified laboratory possessing the adequate equipment to carry out their synthesis and subsequent purification, should handle highly toxic chemical warfare agents. The Forensic Science Center at Lawrence Livermore National Laboratory (LLNL) has the authority and capability to synthesize and handle small quantities of GF, GD or VX through its accreditation as a United States Designated Laboratory for the Organization for the Prohibition of Chemical Weapons, which performs monitoring for verification of international treaties that ban chemical weapons. Proper protective personal equipment (PPE) should be worn at all times which include lab coats, safety glasses, butyl-based gloves with nitrile gloves underneath to provide further protection and a face shield. The handling and preparation of the NMR samples should be conducted inside a well- ventilated and certified chemical fume hood.

## Molecular dynamics simulation

Molecular dynamics (MD) simulations were performed with AMBER (version 12) using the recent charges and parameters [50] for 6-OxP-CD and the GAFF force field for the NAs. The program CHIMERA [51] was used to model the NA:6-OxP-CD agent complexation processes. The 6-OxP-CD and the NA:6-OxP-CD complexes were solvated in a box of TIP3P water [52] sufficient in size to have at least 15 Å of water between the solute and the solvent interface ($\sim 51 \times 51 \times 51$ Å$^3$ initial box size) (Fig 3). The systems consisted of about 12500 atoms ($\sim$ 4100 water molecules) and to neutralize the systems, sufficient sodium ions (typically one) were added to it. Each system was energy minimized using 250 steps of steepest descent and 1500 steps of conjugate gradient. Constant temperature and pressure dynamics (NPT) were performed on these minimized systems [53]. Coupling constants of 2 and 5 ps were used for temperature and pressure, respectively. Periodic boundary conditions were used, and electrostatic interactions were treated by particle mesh Ewald methods [54] with a 9 Å cutoff in direct space and a 1 Å grid. Bonds containing hydrogen were constrained using SHAKE [55], and a time step of 2 fs was used in each simulation. The systems were initially coupled to a heat bath at 100 K for the first 100 ps, then increased to 200 K for the next 100 ps, and finally raised to 300 K for the remainder of the simulation. Each simulation was performed for a total of 10 or 30 ns depending on the NA. Some simulations were extended out to 50 ns to investigate the distribution of distances sampled between the NA and 6-OxP-CD in different complex orientations. The first 2.5 ns of the 300 K dynamics were used for equilibration.

The free energies of binding between 6-OxP-CD and the three NAs (e.g., GF, GD and VX) were estimated using the Molecular Mechanics-Generalized Born surface area (MM-GBSA) method from snapshots of the solvated trajectories [56–58] MM-GBSA energy calculations were performed on replicate simulations and then averaged to obtain the average binding energy for a particular NA:6-OxP-CD system (Fig 3). The binding free energy was estimated by the equation:

$$\Delta G_{binding} = G_{complex} - (G_{6-OxP-CD} + G_{NA})$$

where each term, G is estimated as the sum of gas-phase molecular mechanics energy $E_{gas}$ and

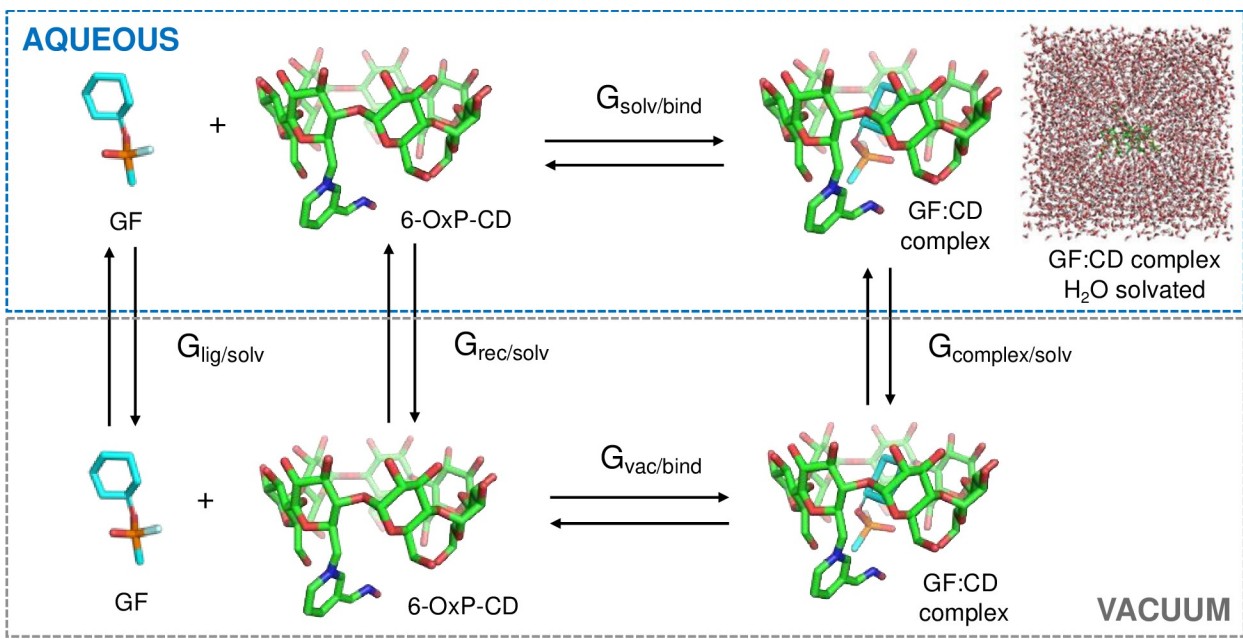

**Fig 3.**

the solvation energy,

$$G_{total} \ = \ E_{gas} \ + \ G_{sol}$$

The contribution of entropy (S) was neglected during these free energy calculations. The solvation free energy ($G_{sol}$) is the sum of the polar and nonpolar solvation energies of the molecules determined by solving the Generalized Born (GB) equation. The binding free energies for the complexes were calculated using the MMPBSA.py script [59] in AMBER12 on snapshots from each 7.5 ns trajectory. The modified Onufriev-Bashford-Case-I GB (ib = 2) model [60, 61] was used for the calculation with a NA concentration of 0.24 mM. The surface tension used to calculate the nonpolar contribution to the free energy of solvation was 0.0072 kcal mol$^{-1}$ Å$^{-2}$.

## Results and discussion

Inclusion complexes mediated by CDs as hosts have been extensively studied and have found numerous applications in the pharmaceutical field mainly in the form solubility enhancers of small molecule therapeutics [62, 63]. It is this inclusion complex formation ability that now is experiencing a sudden rebirth in the CWA medical countermeasures (MCMs) area since the inactivation of certain NAs by native CDs was first described back in the mid-80s [38, 39]. Since then, landmark, collaborative work between the Estour group at the Université de Rouen in France and the Worek group at BIPT in Germany has yielded more structurally elaborate CDs, designed specifically to be more efficient at binding and degrading NAs [64–66]. Most of the CDs produced with the goal of degrading NAs were not specifically designed to be used as MCM scaffolds but rather as components in the preparation of NA-hydrolyzing solutions and as such were not extensively tested in biological assays. However, in 2011 the Kubik group at the Technische Universität Kaiserslautern in Germany reported the use of 6-OxP-CD, an oxime-bearing CD, in the efficient degradation of GF [40]. Although their

studies solely involved GF and no other NAs, it represented the first time that a CD construct of this kind was being evaluated as a potentially useful MCM. These results attracted our attention and as the original report lacked a desirable experimental or computational modeling component showing the nature of the GF:6-OxP-CD inclusion complex to explain such efficient degradation. Therefore, in an effort to identify structural features that make 6-OxP-CD such an efficient scaffold for the degradation of GF, a detailed computational analysis on this system was performed. This work then expanded into the study of inclusion complexes between 6-OxP-CD and two additional NAs, GD and VX, that were not screened in the original work.

The ability of 6-OxP-CD to degrade GF was demonstrated by the Kubik group at the BIPT, and this function was evaluated by using the Ellman's assay to quantify AChE's activity [67]. 6-OxP-CD was tested only against GF and its activity against other nerve agents, even those belonging to the G-series was never reported. After studying the work performed by the Worek group, we found it 6-OxP-CD's activity against GF appealing and interesting, and thus decided to evaluate its activity against two other nerve agents of interest, namely Soman (GD) and VX. Our experimental setup for these evaluations is different from the one originally published and differs in several facets outlined in Table 1. The first one is the experimental readout for the activity of the CD construct, whereas the original work used the Ellman's assay to measure the protective activity of 6-OxP-CD we decided to use $^{31}$P-NMR for this purpose. Analysis of the reaction mixture by $^{31}$P-NMR allows us to directly observe the rate of nerve agent degradation (if it occurs) and also to determine if the formation of a CD:nerve agent inclusion complex is formed during this process. Our use of $^{31}$P-NMR spectroscopy to follow the degradation of GF constitutes a direct measurement relative to the Ellman's assay which is an indirect measurement.

A second element of comparison between the two methods is the nature of the buffer used. The buffer Tris(hydroxymethyl)aminomethane hydrochloride (TRIS-HCl) was used in the original method while we opted for 3-(*N*-morpholino)propanesulfonic acid (MOPS) for our studies. The reason for employing MOPS was to minimize GF background degradation which is a well-known event when using TRIS-HCl and commonly used phosphate buffers [68] and thus allowing us to get a more accurate readout of the effect of adding the 6-OxP-CD in the mixture. Another element of comparison deals with the substrate concentrations used in both studies. At BIPT, the highest concentration of GF used was 500 μM in their *in vitro* assays while in our $^{31}$P-NMR experiments, the nerve agent concentrations are an order of magnitude higher (~5000 μM). The last point of comparison deals with the ratio of 6-OxP-CD to GF used in both experiments. At BIPT, the researchers used a large excess of the 6-OxP-CD to the nerve agent (500:1 and 50:1). For our NMR studies here, we evaluated the 6-OxP-CD at a 1:1 stoichiometric ratio to GF and also evaluated its possible use in catalytic fashion, that is 10% of the total nerve agent concentration.

**Table 1. Comparison of the evaluation methods for determining 6-OxP-CD's ability to degrade GF at the Bundeswehr Institute for Pharmacology and Toxicology (BIPT) and the Forensic Science Center (FSC) at LLNL.** While studies at BIPT focused on GF only, our studies have expanded the evaluation of 6-OxP-CD to include GD and VX.

|  | **Zengerle *et al.* (2011) (BIPT)** | **This work** |
|---|---|---|
| CD:Nerve agent system | 6-OxP-CD:GF | 6-OxP-CD:GF/GD/VX |
| Experimental readout | Ellman's AChE reactivation assay | $^{31}$P-NMR spectroscopy |
| Buffer system/Temperature | TRIS-HCl (pH 7.4) / 37 °C | MOPS (pH 7.2) / 37 °C |
| Nerve agent concentration | 500 mM | 5000 mM |
| Stoichiometry (CD:NA) | 500:1; 50:1 | 1:1; 0.1:1 |

Our experimental design involved testing of 6-OxP-CD and its isolated components to evaluate whether or not the covalent association of these is the origin of the significant degradative properties above background. To this end, we added to our panel for evaluation native β-CD and 3-pyridine aldoxime (3-PA) in addition to the control run involving background nerve agent degradation in the MOPS buffer (pH = 7.2). Our studies rely on the use of [31]P-NMR to directly follow the degradation of GF in the buffer as well as in the presence of 6-OxP-CD, native β-CD and 3-PA. Fig 4A shows the [31]P-NMR spectra for GF at the outset of the experiment (t = 5 min.) in the buffer. There are two signals that comprise the full spectrum, a doublet centered at δ = 33.9 ppm for GF with a coupling constant of 1045 Hz. The other signal arising from a species in low concentration is also a doublet centered at δ = 28.3 ppm with a coupling constant of 966 Hz belonging to methylphosphonofluoridic acid, a commonly observed degradation product of GF under acidic conditions.

Background hydrolysis of GF in the buffer is slow and even after 24 hours ~60–65% of the nerve agent remains in the mixture, with the remaining [31]P-NMR signals belonging to that of CMPA (δ = 25.6 ppm) and methylphosphonofluoridic acid (doublet, δ = 28.3 ppm) (Fig 2B and 2E). In similar fashion, incubation of GF with a stoichiometric amount of 3-PA resulted only in partial degradation of the nerve agent and after 24 hours ~60% of the agent remains in the buffer (Fig 4C). Therefore, there is no degradative effect brought upon by the addition of 3-PA to the buffer system, however, an important long-lived phosphorus-containing species can be observed as a singlet centered at δ = 39.1 ppm. Incubation of GF with an equimolar amount of 6-OxP-CD at pH 7.2 and 37 °C results in the immediate degradation of the nerve agent (Fig 4D and 4F). The diagnostic doublet belonging to GF cannot be detected in the first spectrum obtained for the mixture (t = 5 min.) demonstrating the efficiency of the CD at rapidly degrading this particular agent. In fact, it can be observed that after this short time, most of the GF has been converted to CMPA, in addition to other phosphorus-containing species denoted with the blue diamonds (Fig 4B).

An interesting finding that was observed during these initial rounds of [31]P-NMR experiments with GF was the observation that native β-CD itself forms a stable complex with the nerve agent. Fig 5 shows the immediate formation of the GF:β-CD complex by the marked upfield shift of the doublet assigned to GF, from δ = 33.9 ppm to an initially apparent "doublet of doublets" centered at δ = 33.4 ppm (Fig 5A and 5B). The chemical shift change of NMR signals for the [31]P nucleus upon forming an inclusion complex with a CD host is a known phenomenon and its direction, upfield versus downfield, correlates to the positioning of the nucleus in the interior of the CD cavity [69]. In our experiment, this chemical shift change is observed and the inclusion complex appears to be long lasting but the overall rate of hydrolysis of the agent is similar to the one observed for the degradation in buffer alone. An interesting observation that supports the formation of a GF:β-CD inclusion complex is the fact that in addition to the upfield shift of the signal is the apparent splitting of each signal into what appears to be two separate doublets with small coupling constants. However, these are actually two overlapping doublets still retaining the large coupling constants associated with a P-F bond ($J$ = 1045 Hz each). This phenomenon arises from the enantiomeric resolution of both GF isomers ($R$ and $S$ at the phosphorus center) when the nerve agent enters the chiral interior of β-CD. The formation of this inclusion complex between β-CD and GF explains the protective capacity that this CD has on AChE from the effects of the nerve agent [43]. As shown in the [31]P-NMR spectra in Fig 3C, there is a rapid β-CD:GF inclusion complex formation (t = 5 min.) resulting in the two doublets arising from each GF enantiomer in the interior of the CD (δ = 33.4 and 33.3 ppm) and the formation of a singlet signal centered at δ = 35.7 ppm whose identity is currently under investigation, and it can be observed that as time progresses it grows in magnitude just as the signal for CMPA (δ = 24.9 ppm) does.

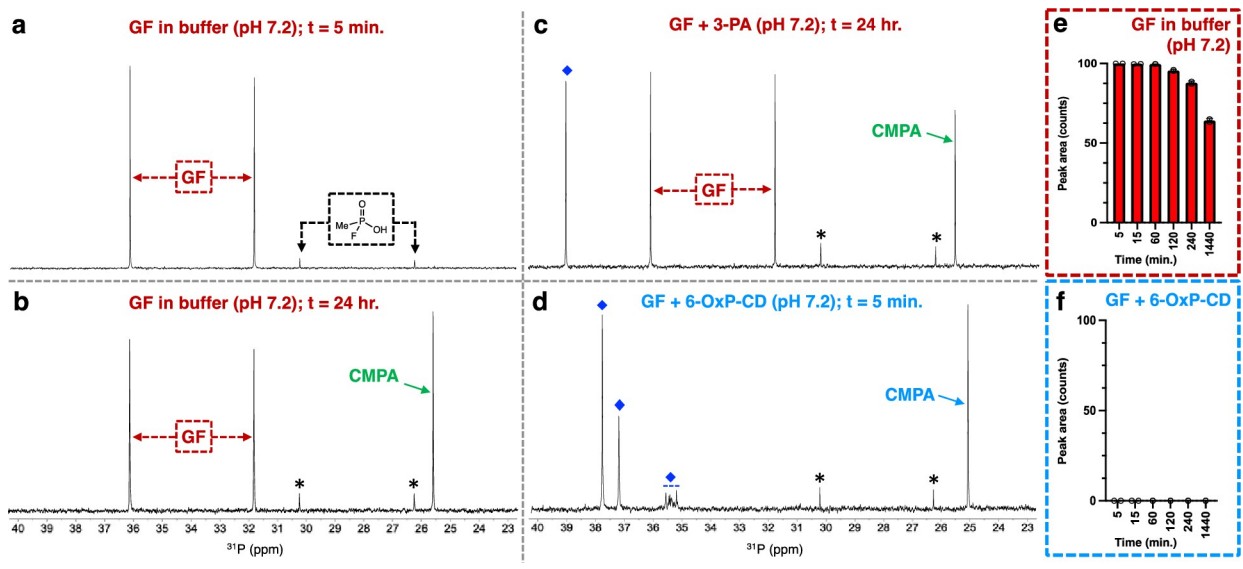

**Fig 4.**

Next, we turned our attention to Soman (GD) and evaluated the ability of 6-OxP-CD, β-CD and 3-PA to degrade this G-series nerve agent that poses the greatest risk among these due to its fast aging once it adducts AChE (aging $t_{1/2}$ = 2 min.) [1, 70]. Fig 6A shows the $^{31}P$-NMR of GD at the outset of the experiment (t = 5 min.) in the buffer at 37 °C. The spectrum is dominated by the two sets of doublets belonging to each of the two GD diastereomers centered at δ = 34.4 ppm and 33.7 ppm each and featuring a coupling constant of $J$ = 1045 Hz. Prolonged

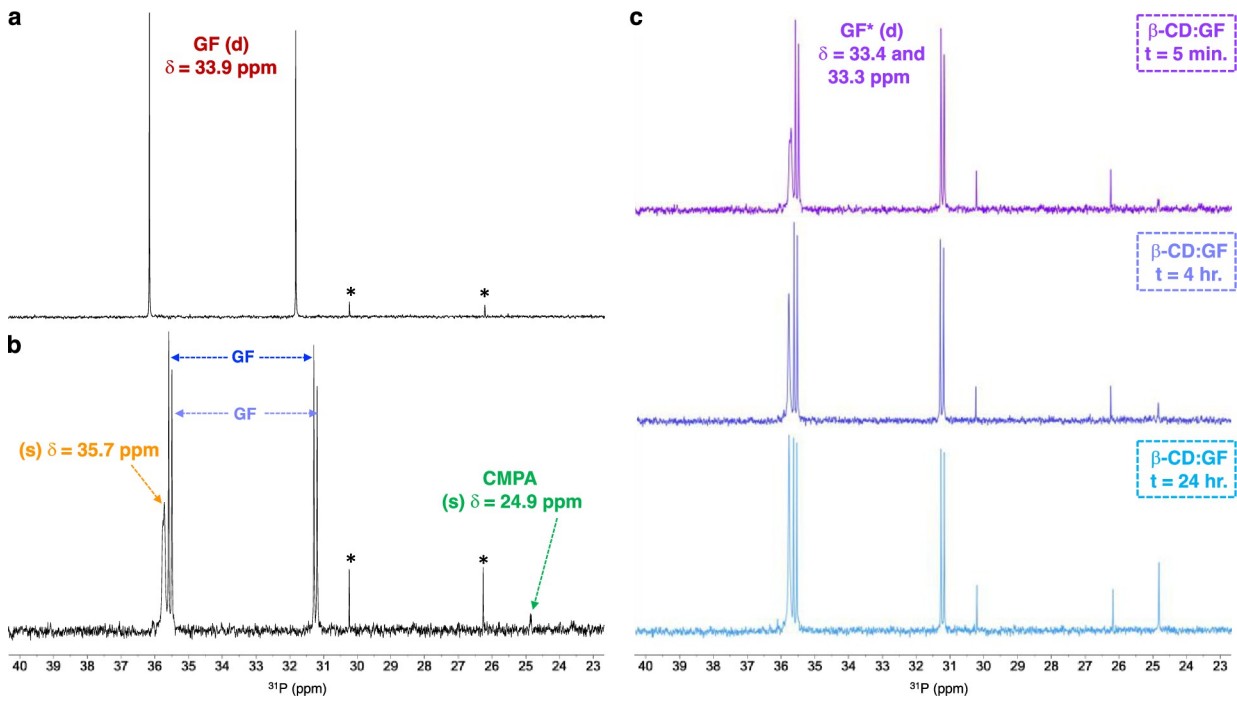

**Fig 5.**

incubation of GD in the buffer over a 24-hour period of time shows that the nerve agent slowly degrades to two products, one of them fluoromethylphosphonic acid (doublet centered at δ = 28.3 ppm; $J$ = 966 Hz) and pinacolyl methylphosphonic acid (singlet centered at δ = 25.2 ppm) (Fig 6B and 6E). In contrast to its rapid degradation of GF, 6-OxP-CD does not degrade GD with the same rapidity and accomplishes this task in an overnight basis at 37 °C and pH 7.2 (Fig 6C, 6D and 6F). From the $^{31}$P-NMR spectra shown in Fig 6C, it can be observed that a 6-OxP-CD:GD inclusion complex is immediately formed, and the degradation of the nerve agent begins to take place at a faster rate than background buffer hydrolysis (Fig 6E). Using the $^{31}$P-NMR data gathered by integrating the signals arising from the nerve agent and following their gradual disappearance, we can calculate that full degradation of GD, assuming first-order kinetics, occurs at t ~ 303 min (~5 hours) (Fig 6F). A hypothesis, supported by our computational modeling on the 6-OxP-CD:GD complex (vide infra), is that binding of GD occurs in an orientation whereby the P-center is not in close proximity to the oxime moiety. Given that this binding event is dynamic, the next time GD enters the interior of 6-OxP-CD it does so in an orientation that now facilitates its nucleophilic attack by the oxime moiety. Again, due to the fact that no $^{31}$P-NMR signals arising from free GD were observed, it can be anticipated that this nerve agent is fully neutralized and as such should not pose a toxic threat to AChE in the event of an exposure. Furthermore, in the $^{31}$P-NMR experiments belonging to the 6-OxP-CD:GD system, we observed the appearance of numerous phosphorus-containing species that have been proposed as CD-nerve agent covalent intermediates during the degradation process. The appearance of such species and structure determination by LC-MS analysis was previously reported for the 6-OxP-CD:GF system (Fig 6C and 6D).

Our last test for 6-OxP-CD involved the experiments with the nerve agent VX. Structurally and physically, VX is very different to GF and GD and as such was expected to behave differently in our set of experiments. One important difference between VX from the G-series agents used in this work lies in the nature of the leaving group, which is the N,N-diisopropylaminoethylthiolate anion versus the fluoride ion in GF or GD. Therefore, the leaving group in VX is much larger and it was anticipated that it could have a deleterious effect on its overall binding to the CD cavity in 6-OxP-CD as well as β-CD if such complex formation was possible. Since VX does not have a P-F bond, its $^{31}$P-NMR signal is a singlet centered at δ = 61.0 ppm (Fig 7A). It can be observed that VX in plain buffer at 37 °C slowly degrades and after a 24-hour period of time ~37% of the original nerve agent still remains in the medium (Fig 7B and 7E). The other components in the mixture can be attributed to O-ethyl methylphosphonothioate (δ = 75.6 ppm), the equally toxic by-product EA-2192 (δ = 42.5 ppm) [71] and the non-toxic ethyl methylphosphonate (EMPA, δ = 26.7 ppm). The signal occurring at δ = 63.1 ppm (denoted with an * in Fig 7A) is an impurity in the starting material used in the synthesis of VX. Incubation of 6-OxP-CD with VX, results in a markedly faster degradation rate of the nerve agent and within five minutes into the reaction, one can observe the appearance of signals belonging to EA-2192 and EMPA (Fig 7C). The degradation of VX continues under these conditions until ~20% of the nerve agent remains in the medium after 24 hours of reaction time (Fig 7D and 7F). However, unlike the cases above with GF and GD, no tight inclusion complex between 6-OxP-CD and VX is observed by $^{31}$P-NMR indicating that the nerve agent is not fully sequestered by the CD's interior (vide infra). Although, the collected data on the 6-OxP-CD:VX system may not be very promising, the observation that VX degrades faster in the buffer when 6-OxP-CD is present (in an equimolar ratio) demonstrates that the 6-OxP-CD is active to some extent and that it is not the optimal CD scaffold for this specific nerve agent. In fact, 6-OxP-CD could be used as starting platform for further structure-activity relationship (SAR) studies to increase its binding and hydrolytic activity against VX.

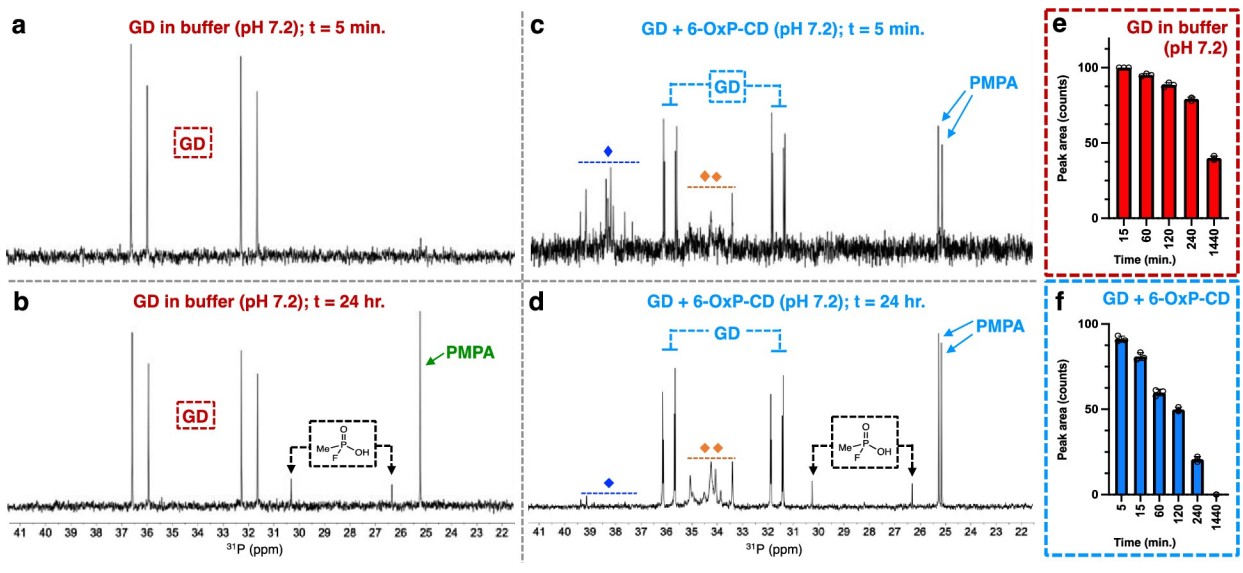

**Fig 6.**

Incubation of the nerve agents GD and VX with β-CD and 3-PA showed similar degradation profiles during the $^{31}$P-NMR experiments. Thus, incubation of GD in the presence of β-CD results in the slow degradation of the agent with a similar profile to that of its degradation in buffer only after 24 hours (Fig 8A). As observed with GF, β-CD forms an inclusion complex with GD. Thus, it can be anticipated that this binding event by β-CD effectively neutralizes the nerve agent from interacting with AChE in similar fashion to the GF case. The same degradation results are obtained when GD is incubated with 3-PA, with ~60% of the original agent degraded after a 24-hour period of time (Fig 8B). Similarly, incubation of β-CD with VX results in the slow degradation of the nerve agent and curiously at a much slower rate than plain buffer (Fig 8C). At the end of the 24-hour period of time ~80% of the VX agent still

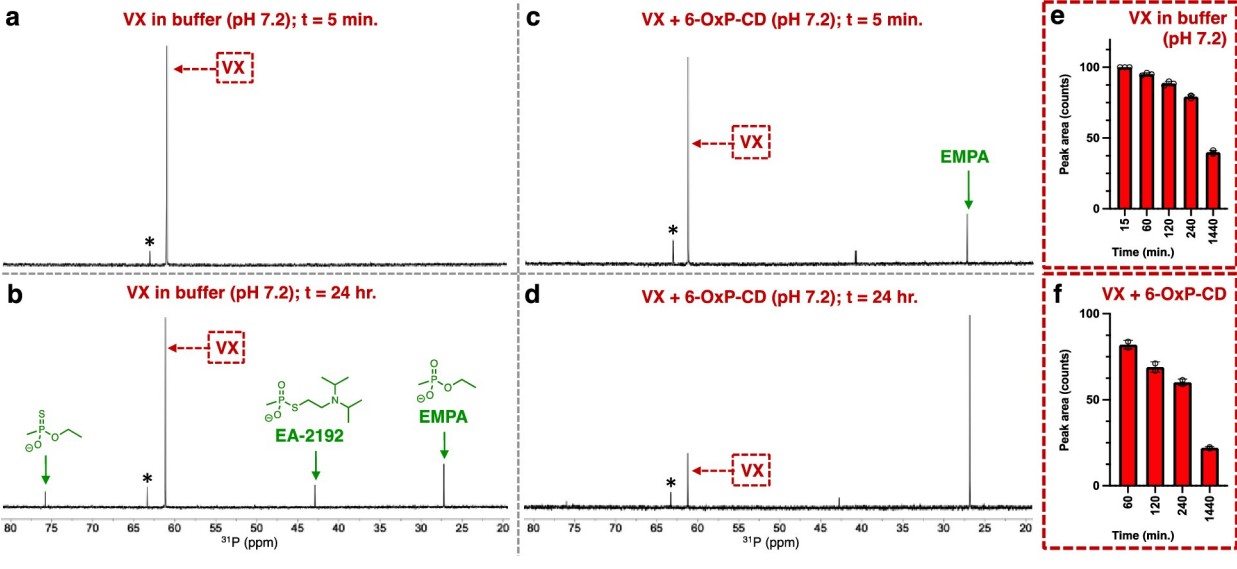

**Fig 7.**

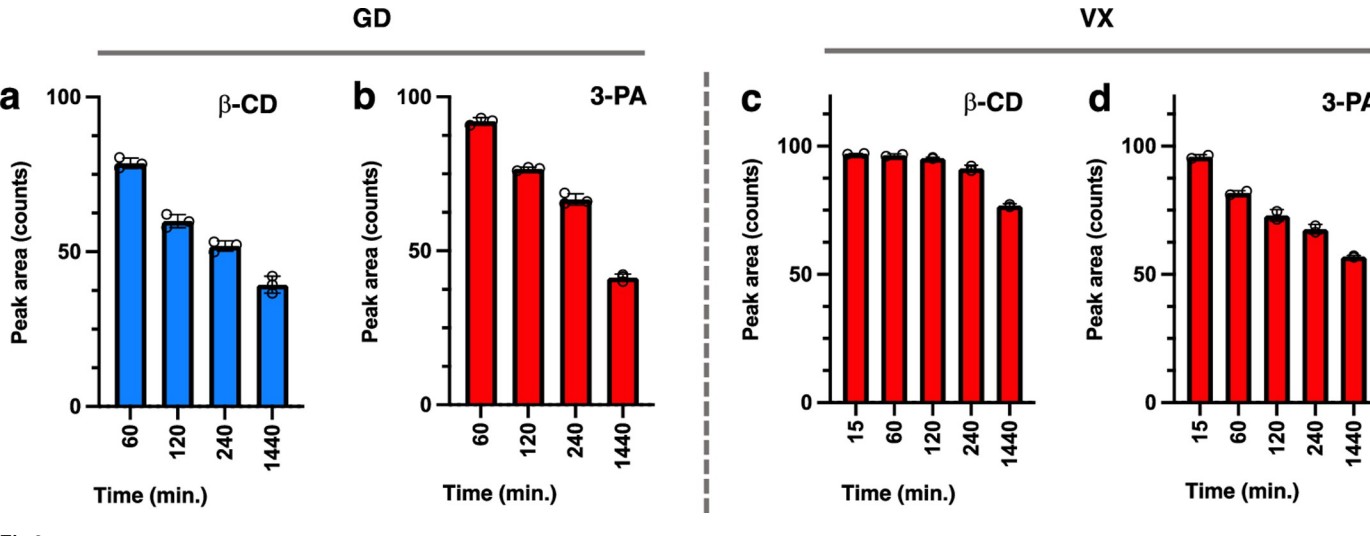

**Fig 8.**

remains in the mixture in free form. It is hypothesized that VX does bind to β-CD in a very weak manner as predicted by computational modeling on this system, and this weak binding still serves to confer some shielding of the phosphorus center from attack by the hydroxide ions in solution. Incubation of 3-PA and VX shows that there is no significant degradation of VX with the added oxime and the results are very similar to the ones observed in the plain buffer (Fig 8D).

## Simulations on the GF:6-OxP-CD complex

The first inclusion complex studied via MD simulations was the one between GF and 6-OxP-CD. The complex was initially chosen as it was the first and only one published by the Kubik group [40]. Despite the evaluation of 6-OxP-CD against GF there were no reports on its ability to bind and hydrolyze other NAs. MD simulations were carried out using the program AMBER while quantum mechanical (QM) calculations were performed using Gaussian in order to create parameters for the 6-OxP-CD. The oxime was modeled in the ionized form in 6-OxP-CD and calculated structures were used to create AMBER parameters using the program Antechamber while the partial charges were obtained from AM1-BCC calculations. A notation that will be used throughout the manuscript to describe the nature of the inclusion complex for all three agents with 6-OxP-CD separately originates from the manner in which the agent is 'introduced' into the 6-OxP-CD through its wide, $2^o$ rim. Hence, the 'up' conformation corresponds to introducing the agent into the CD cavity through the wide rim of 6-OxP-CD with the P = O moiety pointing *away* from the interior of the CD (i.e., the P = O is in contact with the exterior of the CD), and this is denoted as 'up-$_{NA}$'. In the 'down' conformation, the agent is introduced into 6-OxP-CD with the P = O moiety oriented *towards* the interior of the CD (i.e., the P = O is in close proximity to the $1^o$ rim) and this is denoted as 'down-$_{NA}$. Thus, using these notations and established computational parameters for the modeling, GF was introduced in the cavity of the model 6-OxP-CD in these two orientations, denoted 'up-$_{GF}$' and 'down-$_{GF}$' (Fig 9). In the 'down-$_{GF}$' complex conformation, GF is positioned so that the phosphorus moiety lies in the vicinity of the narrow $1^o$ rim of 6-OxP-CD (e.g., towards the C6 hydroxyl groups) where coincidentally the pyridinium aldoxime moiety is located (Fig 9A). In this 'down-$_{GF}$' complex conformation, the large cyclohexyl moiety of the

agent faces the wide 2° rim of 6-OxP-CD (e.g., towards the C2 and C3 hydroxyl groups). In the 'up-$_{GF}$' complex conformation, GF is positioned with its P = O moiety facing towards the wide 2° rim of 6-OxP-CD while its cyclohexyl group lies buried in the interior of 6-OxP-CD (Fig 9B). Interestingly, during the simulations of this 'up-$_{GF}$' complex conformation, the pyridinium aldoxime moiety was found to occasionally pack in parallel fashion to the narrow rim in a way that resembles a lid at this opening. It is not known if there is any functional significance to this conformation.

Simulations using the charged oxime (N-O⁻) demonstrated that both conformations (i.e., 'up-$_{GF}$' and 'down-$_{GF}$') were very stable ('down-$_{GF}$' is slightly more stable from the MM/GBSA energies, see S6 File) and one of the first line of calculations conducted involved measuring the distance between the phosphorus atom of GF and the anionic oxygen of the oxime in 6-OxP-CD (i.e., $P_{GF}\cdots O_{Oxime}$) throughout the time of the simulation (t = 50 ns). Representative plots for these changing $P_{GF}\cdots O_{Oxime}$ distances are shown in Fig 5. In the 'down-$_{GF}$' simulation, the $P_{GF}\cdots O_{Oxime}$ distance was found to be ~ 4–5 Å for time periods of up to 5 ns (Fig 10A). These observations suggest that there exists more than enough time for the oxime to find an orientation that is in close enough proximity to attack the P center in GF and cause its degradation. In contrast, during the 'up-$_{GF}$' simulation, the $P_{GF}\cdots O_{Oxime}$ distance lies most of the time between 6–8 Å and up to > 10 ns at a time making this orientation an unlikely model to explain 6-OxP-CD's unparalleled degradative capacity towards GF (Fig 10C). It is in this orientation that $P_{GF}\cdots O_{Oxime}$ distances lying between 12–14 Å can be observed with time intervals of ~3–4 ns establishing that this inclusion complex's conformation could not be effective at degrading GF. An additional set of comparative data analyses between the 'up-$_{GF}$' and 'down-$_{GF}$' complex conformations involved measuring the distances between the center of mass (COM) of the 6-OxP-CD and the center of mass of GF. This was performed to evlauate if there is an additional clear link, aside from directly measuring $P_{GF}\cdots O_{Oxime}$ distances, that would help explain the high reactivity of 6-OxP-CD towards GF. The distances between the COMs of both components were found to be consistently smaller in the 'down-$_{GF}$' (i.e., ~ 0–1.5 Å) orientation (Fig 10B) than the 'up-$_{GF}$' orientation (i.e., ~ 2.5–4.0 Å) (Fig 10D), supporting the fact that the GF lies deep in the cavity and in close proximity to the oxime in the 'down-$_{GF}$' conformation.

## Simulations on the GD:6-OxP-CD complex

The modeling of GD with 6-OxP-CD shows a similar pattern to the one observed for GF however, it appears that due to its geometry and more degrees of freedom at the pinacolyl moiety, GD adopts conformations in the interior of 6-OxP-CD that are not ideal for nucleophilic attack by the oxime. In the 'down-$_{GD}$' conformation, GD is positioned so that the phosphorus moiety lies in the vicinity of the narrow 1° rim of 6-OxP-CD (e.g., towards the C6 hydroxyl groups) and therefore close to the oxime (Fig 11A). In this 'down-$_{GD}$' conformation, the large pinacolyl moiety of the agent faces the wide 2° rim of 6-OxP-CD (e.g., towards the C2 and C3 hydroxyl groups) and as in the GF case, it is buried in the interior of the CD cavity. In the 'up-$_{GD}$' conformation, GD is oriented with its phosphorus center towards the wide 2° rim of 6-OxP-CD while its pinacolyl group lies in the interior of the 6-OxP-CD cavity (Fig 11B). The MM/GBSA energies show that the 'down-$_{GD}$' and 'up-$_{GD}$' conformation are identical (S6 File).

Representative plots for the constantly changing $P_{GD}\cdots O_{Oxime}$ distances are shown in Fig 12. In the 'down-$_{GF}$' simulation, the $P_{GD}\cdots O_{Oxime}$ distance was found to be ~ 4–6 Å for time periods of up to 16–18 ns (Fig 12A). These observations suggest that there exists more than enough time for the oxime to find a conformation that is in close enough proximity to attack the P center in GD and degrade it, even with a higher probability than in the GF case (t ~ 5

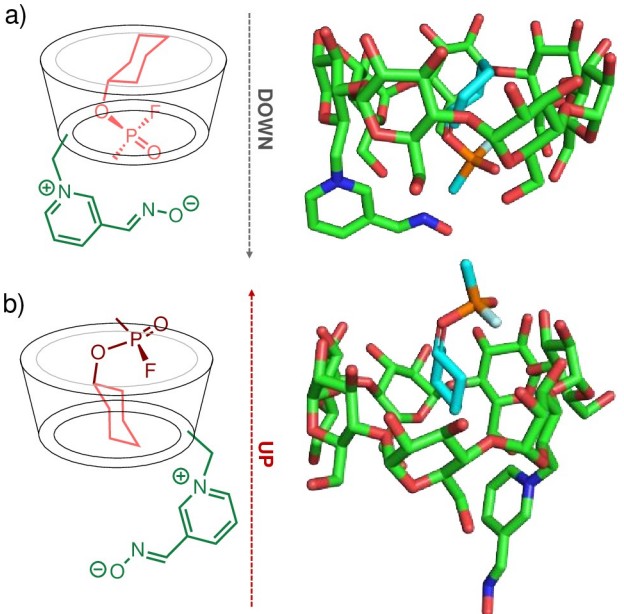

**Fig 9.**

ns). Interestingly, during the 'up-$_{GD}$' simulation, the $P_{GF}\cdots O_{Oxime}$ distance lies between 12–15 Å which effectively makes this conformation an unlikely model to explain 6-OxP-CD's degradative capacity towards GD (Fig 12C). It is in this conformation that $P_{GF}\cdots O_{Oxime}$ distances lying beyond 11 Å can be observed with time intervals of > 40 ns highlighting its inability to be of any use in GD degradation. Analysis of the COM data for the GD:6-OxP-CD complex shows very similar results to those found for the GF:6-OxP-CD complex. The distances

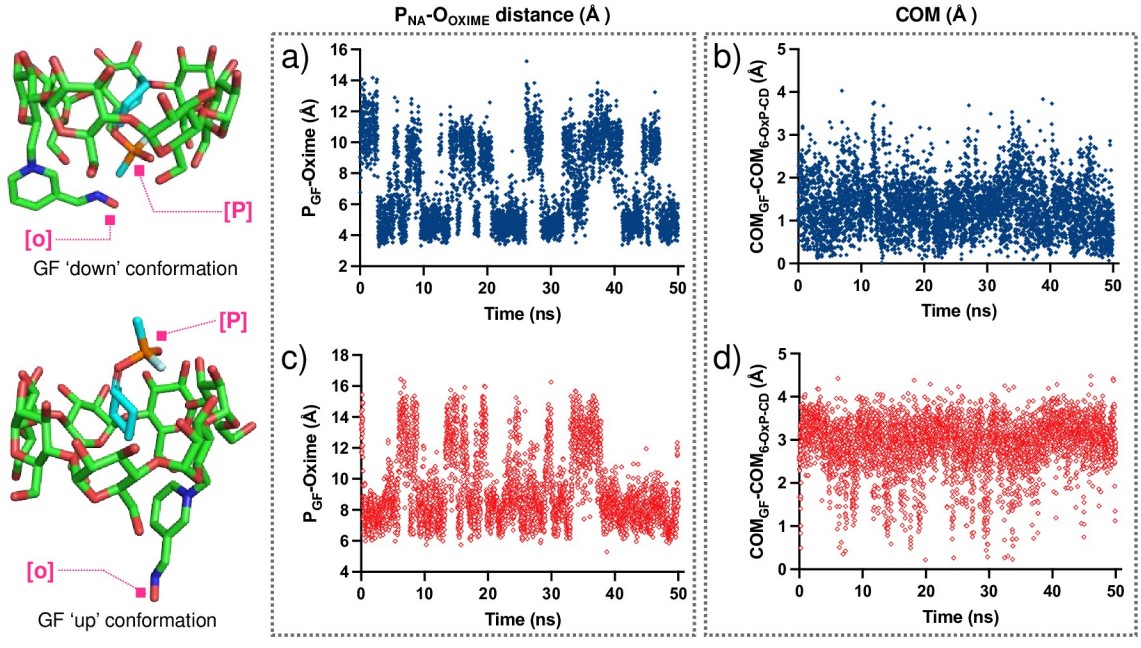

**Fig 10.**

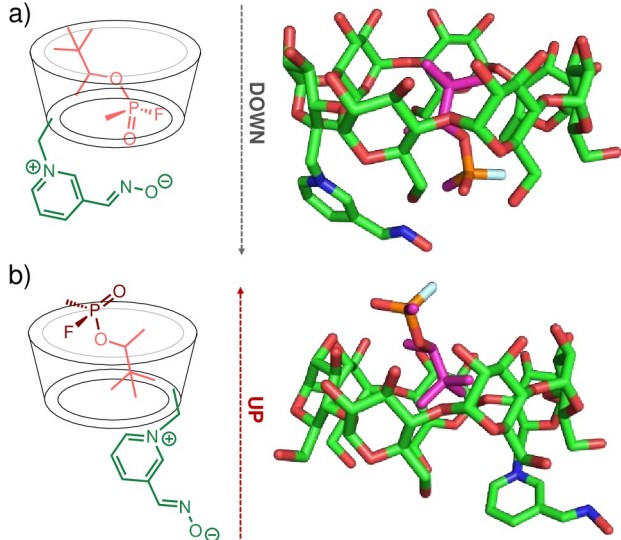

**Fig 11.**

between the COMs of both components were found to be consistently smaller in the 'down-$_{GD}$' (i.e., ~ 0–1.5 Å) conformation (Fig 12B) than the 'up-$_{GD}$' conformation (i.e., ~ 2.0–4.0 Å) (Fig 7D), again supporting the fact that the phosphorus center lies in closer proximity to the oxime in the 'down-$_{GD}$' conformation.

## Simulations on the VX:6-OxP-CD complex and VX:6-OxP-α-CD

Modeling of VX in the interior of 6-OxP-CD did not suggest the formation of a stable inclusion complex. It was found that in only 1 out of the 20 simulations performed the VX molecule

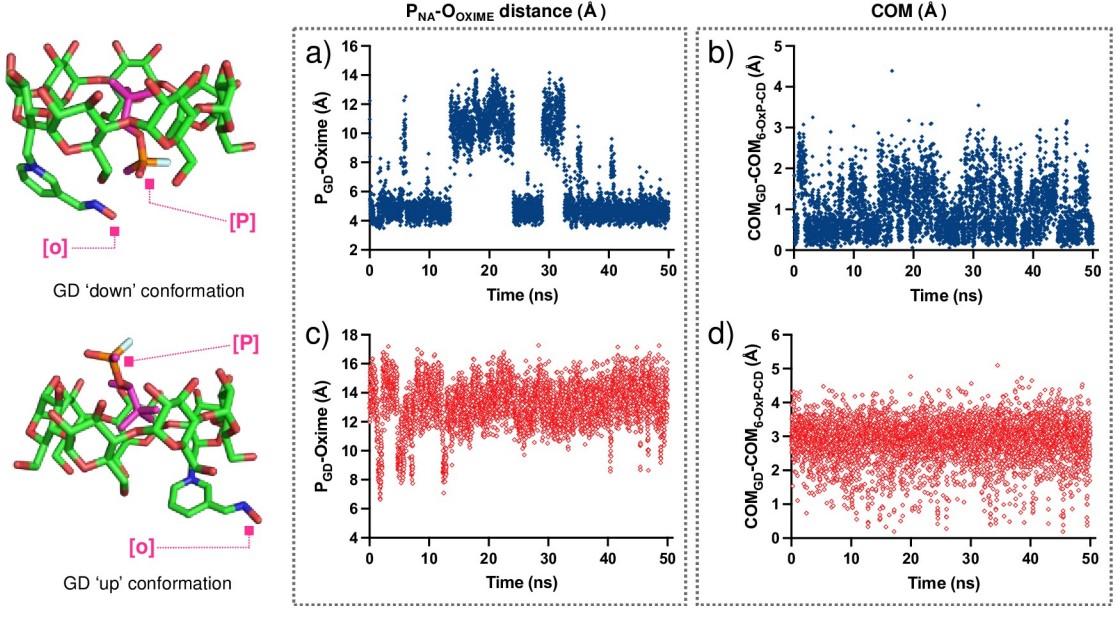

**Fig 12.**

tended to remain in the interior of 6-OxP-CD and suitably positioned itself for attack by the oxime nucleophile. The remaining simulations strongly indicated that VX's diisopropylamino side chain tends to occupy the interior and that this orientation naturally forces the phosphorus center away from the cavity and consequently away from the oxime (Fig 13A). To this end, calculations using a much smaller version of 6-OxP-CD, namely 6-OxP-α-CD were undertaken and found that this α-CD-based construct might be a more suitable host for VX. In similar fashion tot eh VX:6-OxP-CD system, with the 'up-$_{VX}$' conformation, it was found that the VX remained in the interior of the 6-OxP-α-CD, in only 1 of 20 simulations, while it preferred to leave the cavity in the remaining simulations. However, when VX is modeled in the 'down-$_{VX}$' conformation with 6-OxP-α-CD, the VX remained in the interior of the CD where the phosphorus center lies close to the $2^o$ rim and quite far from the oxime oxygen in 6 out of 10 simulations (Fig 13B).

Plots for the varying $P_{VX}\cdots O_{Oxime}$ distances when the VX is in the 'down$_{VX}$' orientation, are shown in Fig 14. In all these cases, the phosphorus center of VX is found to lie far away from the oxime oxygen ($P_{VX}\cdots O_{Oxime}$ ~ 10–13 Å) during a significant part of the simulation (t ~ 10–12 ns) with some time (t ~ 5 ns) in close proximity to the oxime ($P_{VX}\cdots O_{Oxime}$ ~ 6–7 Å) (Fig 13A). Analysis of the COM data in the VX: 6-OxP-α-CD complex reveals that the both COMs lie significantly further away ($COM_{VX}$-$COM_{6-OxP-α-CD}$ ~ 4–6 Å) from each other relative to the cases with GF ($COM_{GF}$-$COM_{6-OxP-CD}$ ~ 0.1–2.0 Å) and GD ($COM_{GD}$-$COM_{6-OxP-CD}$ ~ 0.1–1.5 Å) (Fig 14B). The large difference in the ΔCOM values can be explained by the nature of the components for this specific system. In this case, the system involves a much larger NA molecule relative to GF and GD and the much smaller α-CD host. This initial data predicts that although 6-OxP-α-CD might not rapidly degrade VX it can in principle form a stable inclusion complex therefore achieving the protective role required for any medical countermeasure development. The results from the modeling studies between VX and 6-OxP-CD (the $\beta$-CD version) agree with [31]P-NMR experimental data where it was observed that VX never enters the interior of the CD, and its degradation is thus identical to background degradation.

## Conclusions

In this work, we have evaluated the oxime-CD construct 6-OxP-CD for its ability to neutralize the nerve agents GF, GD and VX under physiological conditions. Using [31]P-NMR to follow the degradation of each agent in the presence of 6-OxP-CD, it was found that each possesses a unique degrdative profile. As previously found by the Kubik group, 6-OxP-CD was found to degrade GF immediately, making this scaffold a formidable candidate for use as a medical countermeasure (MCM). With regards to Soman (GD), 6-OxP-CD was found to immediately form an inclusion complex with the agent and although it degraded it at a slower rate than GF ($t_{1/2}$ ~ 2 hours), it still was superior than background degradation of the agent ($t_{1/2}$ ~ 22 hours). In contrast to the G-based agents, 6-OxP-CD was not observed to form an inclusion complex with VX, with the nerve agent's degradation profile similar to that of background ($t_{1/2}$ ~ 24 hours). In addition to testing 6-OxP-CD, we included its separate structural components, namely β-CD and 3-pyridine aldoxime (3-PA) and evaluated them in their ability to degrade all three agents. We found that β-CD formed stable inclusion complexes with GF and GD, effectively neutralizing them while not forming one with VX. To seek an understanding of the relative activity of 6-Oxp-CD with each nerve agent analyzed in these studies, Molecular Dynamics (MD) simulations coupled with Molecular Mechanics-Generalized Born surface area (MM-GBSA) calculations were applied to the study of 6-OxP-CD, an oxime-bearing cyclodextrin, and its inclusion complexes with the nerve agents GF, GD and VX. The

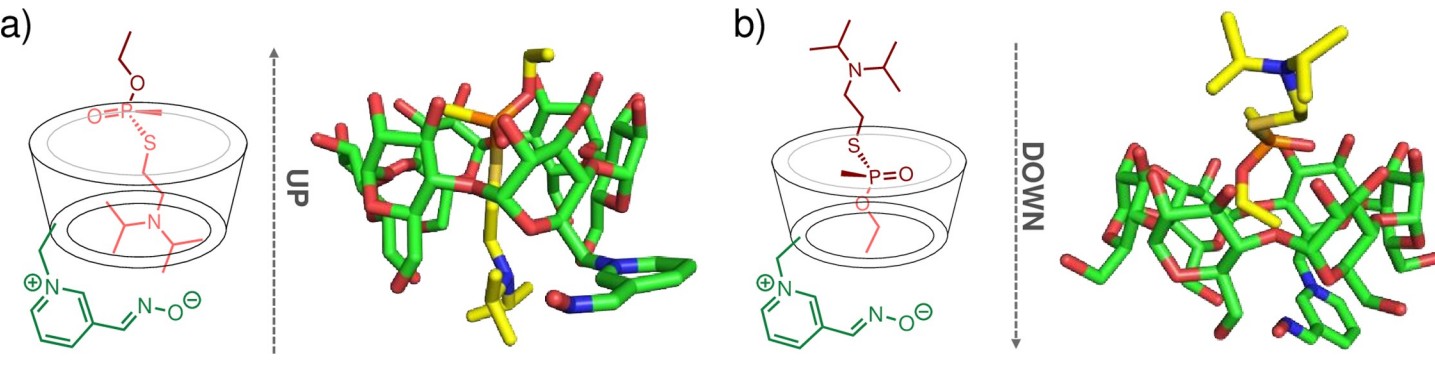

**Fig 13.**

computational studies have helped understand the different behavior of 6-OxP-CD with each one of these nerve agents. In the GF:6-OxP-CD system, when the agent adopts a 'down' conformation in the interior of the CD, the short distance between the oxime oxygen and the agent's phosphorus center can be observed ($P_{GF} \cdots O_{Oxime} \sim 4$ Å). In the 'up' configuration, the distance between the oxime oxygen and the agent's phosphorus center is larger in magnitude ($P_{GF} \cdots O_{Oxime} \sim 6$–7 Å), which suggest that this mode of binding, even though energetically favored, is likely not one that leads to efficient degradation of GF by 6-OxP-CD. Evaluation of GD and VX showed some marked differences from the GF:6-OxP-CD system. For instance, experimental results in our laboratory, have shown that although 6-OxP-CD is effective at degrading GD, the kinetics for the hydrolysis are significantly slower than those observed for GF ($t_{1/2} \sim 4$ hr. vs. instantaneous). Modeling the GD:6-OxP-CD complex has shown that the oxime functional group in 6-OxP-CD lies in close proximity ($P_{GD} \cdots O_{Oxime} \sim 4$–5 Å) to the phosphorus center of the nerve agent for most of the simulation when GD is in the 'down-$_{GD}$' conformation, as in the GF case,. However, the 'up-$_{GD}$' conformation forms a stable enough complex to compete with the 'down-$_{GD}$' conformation and it is in this conformation that a large distance is observed ($P_{GD} \cdots O_{Oxime} \sim 12$–14 Å) for most of the simulation, with barely any significant time in a conformation that brings these two species close enough in space for significant degradation to occur ($P_{GD} \cdots O_{Oxime} \sim 8$ Å). Lastly, the modeling of the VX with 6-OxP-CD has shown that there is no appreciable binding between the agent and the CD. In all the calculations, when forced in the 'up-$_{VX}$' or 'down-$_{VX}$' conformations, the agent tended to slide out of the cavity fast enough that significant interactions stabilizing any type of

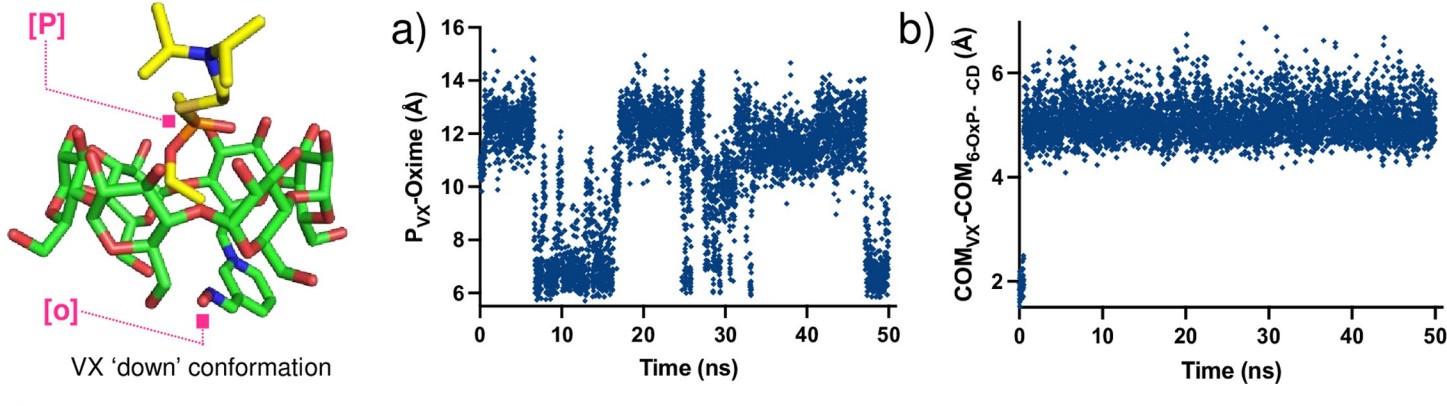

**Fig 14.**

complex that could be useful for degradative activity was not formed. In light of this, further computational studies revealed that VX prefers to adopt the 'down-$_{VX}$' conformation with the ethoxy side chain buried in the interior of the much smaller α-CD version of 6-OxP-CD. In this VX:6-OxP-α-CD complex, the 'up-$_{VX}$' conformation does not lead to the formation of a stable complex, therefore only the data for the down conformation was gathered. In this 'down-$_{VX}$' conformation the phosphorus center lies far away from the oxime oxygen ($P_{VX}\cdots O_{Oxime}$ ~ 10–13 Å) but also spends significant time in close to it ($P_{VX}\cdots O_{Oxime}$ ~ 6–7 Å) anticipating that this smaller 6-OxP-α-CD might not show any rapid degradative properties against VX but could in principle sequester it in its interior, thereby achieving the protective role required for any medical countermeasure development. For medical countermeasure development, a good scaffold is considered to be one that blocks the action of the nerve agent and causes its rapid clearance from the body, while not necessarily causing its degradation. In conclusion, the results presented herein should serve as starting platform to develop more elaborate, powerful cyclodextrin-based hosts with stronger affinities and faster degradative properties against nerve agents other than GF.

## Supporting information

**S1 File. Oxime-phosphorus-distances calculations.**
(XLSX)

**S2 File. Center of masses distances calculations.**
(XLSX)

**S3 File. GF:6-OxP-CD simulation.**
(MPG)

**S4 File. GD:6-OxP-CD simulation.**
(MPG)

**S5 File. VX:6-OxP-CD simulation.**
(MPG)

**S6 File. MM-GBSA molecule topology values.**
(DOCX)

## Author Contributions

**Conceptualization:** Heather A. Enright, Carlos A. Valdez.

**Formal analysis:** Edmond Y. Lau.

**Funding acquisition:** Carlos A. Valdez.

**Investigation:** Edmond Y. Lau, Heather A. Enright, Victoria Lao, Michael A. Malfatti.

**Methodology:** Edmond Y. Lau.

**Resources:** Brian P. Mayer, Audrey M. Williams.

**Software:** Edmond Y. Lau.

**Supervision:** Carlos A. Valdez.

**Writing – original draft:** Carlos A. Valdez.

**Writing – review & editing:** Edmond Y. Lau, Heather A. Enright, Victoria Lao, Michael A. Malfatti, Brian P. Mayer, Audrey M. Williams, Carlos A. Valdez.

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
