## [Decision Letter · Decision Letter 0]

20 Dec 2022

PONE-D-22-30026Molecular Dynamic Simulation Studies on Inclusion Complexes Between 6-OxP-CD, an Oxime-based Cyclodextrin, and the Nerve Agents Cyclosarin, Soman and VXPLOS ONE

Dear Dr. Valdez,

Thank you for submitting your manuscript to PLOS ONE. After careful consideration, we feel that it has merit but does not fully meet PLOS ONE’s publication criteria as it currently stands. Therefore, we invite you to submit a revised version of the manuscript that addresses the points raised during the review process. In your revised manuscript, please pay particular attention to the major concerns raised by Reviewer 2, and endeavour to address them and fully as possible.

We look forward to receiving your revised manuscript.

Kind regards,

Israel Silman

Academic Editor

PLOS ONE

Journal Requirements:

"This work was performed under the auspices of the U. S. Department of Energy by Lawrence Livermore National Laboratory under Contract DE-AC52-07NA27344.  The work was funded by a grant from the Defense and Threat Reduction Agency (DTRA) to C. A. V. (Grant number: CB10902)"

4. We note that you have referenced (ie. Bewick et al. [5]) which has currently not yet been accepted for publication. Please remove this from your References and amend this to state in the body of your manuscript: (ie “Bewick et al. [Unpublished]”) as detailed online in our guide for authors

Reviewers' comments:

Reviewer's Responses to Questions

**Comments to the Author**

1. Is the manuscript technically sound, and do the data support the conclusions?

Reviewer #1: Yes

Reviewer #2: No

2. Has the statistical analysis been performed appropriately and rigorously? 

Reviewer #1: Yes

Reviewer #2: N/A

3. Have the authors made all data underlying the findings in their manuscript fully available?

Reviewer #1: Yes

Reviewer #2: No

4. Is the manuscript presented in an intelligible fashion and written in standard English?

Reviewer #1: Yes

Reviewer #2: Yes

5. Review Comments to the Author

Reviewer #1: This is an interesting paper that uses molecular dynamics to predict the ability of 6-OxP-CD to bind and react with 3 organophosphorus (OP) nerve agents (NAs): GF, GD, and VX. This paper appears to have been stimulated by the observation (ref. 32) that 6-OxP-CD efficiently degraded GF. The computational analysis here rationalized this point, showing close proximity of GF to one orientation (the “down” position) of 6-OxP-CD. The authors then undertook computational analysis of two other OPs, GD and VX. GD appeared to bind 6-OxP-CD in both the “down” and “up” positions. The simulated binding time was quite long in the “up” position where the P atom and the oxime were very far apart. This orientation was proposed to compete successfully with the complex in the “down” position, effectively rationalizing the much slower degradation rate for GD. Additional simulations of VX and 6-OxP-CD failed to show significant binding or reactivity.

The reviewer is not an expert in molecular dynamics simulations, but they seem sound. One comment should be considered for inclusion in the “Conclusions” section:

While not expected in this manuscript, can molecular dynamics be used to make predictions about alternatives to the 6-OxP-CD structure evaluated here? In particular, are isomers available with the 6-Ox moiety or is the CD structure (and its available -OH groups) completely symmetrical? If there is complete symmetry, could one consider a second substituent that would make isomers worth consideration?

Reviewer #2: This manuscript reports the structural analysis of Molecular Dynamics simulations performed for complexes of an oxime-based cyclodextrin (6-OxP-CD) with three nerve agents: cyclosarin, soman and VX. Although the topic has attracted much interest in the literature, gaining insight into the binding and degradation mechanism by derivatized cyclodextrins deserves interest. However, this study seems preliminary and publication at this stage is premature for several reasons.

1) The authors state that they have performed MM-GBSA calculations to estimate the binding free energy. But the results of these computations are not included in the manuscript.

While MM-GBSA calculations could be a valuable ingredient to complement this study, the lack of results (and a detailed discussion about the reliability of MM_GBSA binding affinities) raises serious doubts about some conclusion stated at the end of the manuscript. For instance,

i) regarding the complex with cyclosarin, it is stated that the 'up binding mode', which is not suitable for degradation, is 'energetically favored' relative to the 'down binding mode'. If the difference in binding affinity is substantial, cyclosarin could bind the cyclodextrin, but would never be degraded. How can this be reconciled with the experimental observation that cyclosarin is degraded instantaneously?

ii) regarding the complex with soman, the text states that the 'up binding mode' forms 'a stable enough complex to compete with the 'down-GD' conformation.' Again, without quantitative data, this statement is vague and has little usefulness for the discussion.

2) Several key points of the manuscript are directly related to experimental results obtained by the authors that are not included in the study: reference 34 (unpublished results), which is quoted in pages 6 and 8, the half-life times reported in Conclusions (page 16), or the NMR data collected for assays with VX that support the lack of binding (reported as data not shown in page 15). This is a serious weakness, because the experimental results are fundamental to fully understand the impact of present simulations. In my view, the impact of this study would be much higher if both experimental and computational studies are presented together in a single manuscript.

3) A number of technical details are confusing. For example,

i) while the text states that simulations were performed for a total of 10 or 30 ns depending on the nerve agent (page 7), plots of distances correspond to simulations of 50 ns.

ii) similarly, it seems that replicate simulations were performed for MM-GBSA simulations in order to obtain averaged estimates (page 7), but this is not described subsequently in the manuscript.

iii) more importantly, models were derived for the oxime in ionized and unionized forms (page 9), but results are reported only for the former species. Is the ionized form the most populated in aqueous solution? If not, can the proton be released during the chemical attack to the phosphorus atom? This should be discussed in the text due to the mechanistic implications. Furthermore, the final force field parameters should be provided as Supporting Information.

Overall, the results and conclusions are not fully consistent in this study, which seems to be still preliminary and the significance could be enhanced upon simultaneous inclusion of the experimental data.

6. PLOS authors have the option to publish the peer review history of their article (what does this mean?). If published, this will include your full peer review and any attached files.

Reviewer #1: No

Reviewer #2: No

---

## [Author Response · Author response to Decision Letter 0]

6 Feb 2023

Please see our attached "Point by point responses" document where we have addressed all the referees' questions.

---

## [Decision Letter · Decision Letter 1]

6 Mar 2023

Evaluation of 6-OxP-CD, an Oxime-based Cyclodextrin as a Viable Medical Countermeasure Against Nerve Agent Poisoning: Experimental and Molecular Dynamic Simulation Studies on Its Inclusion Complexes with Cyclosarin, Soman and VX

PONE-D-22-30026R1

Dear Dr. Valdez,

We’re pleased to inform you that your manuscript has been judged scientifically suitable for publication and will be formally accepted for publication once it meets all outstanding technical requirements.

Kind regards,

Israel Silman

Academic Editor

PLOS ONE

Additional Editor Comments (optional):

Reviewers' comments:

Reviewer's Responses to Questions

**Comments to the Author**

1. If the authors have adequately addressed your comments raised in a previous round of review and you feel that this manuscript is now acceptable for publication, you may indicate that here to bypass the “Comments to the Author” section, enter your conflict of interest statement in the “Confidential to Editor” section, and submit your "Accept" recommendation.

Reviewer #1: (No Response)

Reviewer #2: All comments have been addressed

2. Is the manuscript technically sound, and do the data support the conclusions?

Reviewer #1: Yes

Reviewer #2: Yes

3. Has the statistical analysis been performed appropriately and rigorously? 

Reviewer #1: Yes

Reviewer #2: Yes

4. Have the authors made all data underlying the findings in their manuscript fully available?

Reviewer #1: Yes

Reviewer #2: Yes

5. Is the manuscript presented in an intelligible fashion and written in standard English?

Reviewer #1: Yes

Reviewer #2: Yes

6. Review Comments to the Author

Reviewer #1: The authors have responded to Reviewer 1 with a comment about alternative isomers being considered in an earlier publication. My question specifically asked whether molecular modeling can predict whether alternative structures would be good candidates for catalytic hydrolyses. The authors make a comment about the synthesis of additional structures being beyond the scope of the article, and the reviewer agrees. They do not comment on the predictions made from molecular modeling. The reviewer's comment was only a suggestion.

Reviewer 2 requested extensive additions of computational and experimental methods. These additions add clarification to several protocols used by the authors. I (reviewer 1) find them acceptable.

Reviewer #2: I really appreciate the effort of the authors in addressing my criticisms. This is nice contribution and the combined presentation of experimental and computational results yields a solid work.

7. PLOS authors have the option to publish the peer review history of their article (what does this mean?). If published, this will include your full peer review and any attached files.

Reviewer #1: No

Reviewer #2: No

---

## [Editor Report · Acceptance letter]

10 Mar 2023

PONE-D-22-30026R1 

Evaluation of 6-OxP-CD, an Oxime-based Cyclodextrin as a Viable Medical Countermeasure Against Nerve Agent Poisoning: Experimental and Molecular Dynamic Simulation Studies on Its Inclusion Complexes with Cyclosarin, Soman and VX 

Dear Dr. Valdez:

I'm pleased to inform you that your manuscript has been deemed suitable for publication in PLOS ONE. Congratulations! Your manuscript is now with our production department. 

Kind regards, 

on behalf of

Prof. Israel Silman 

Academic Editor

PLOS ONE